🔓 | **Open Peer Review** | Clinical Microbiology | Research Article

# Evidence of high genetic diversity among parasite populations in a schistosomiasis hotspot

Yvonne Aryeetey Ashong,[1,2] Emmanuel Odartei Armah,[3] Jewelna Akorli,[1] Frank Twum Aboagye,[3] Isaac Owusu-Frimpong,[3] Linda Batsa Debrah,[2] Rhoda Lims Diyie,[3] Samuel Armoo,[3] Alexander Yaw Debrah,[4] Mike Yaw Osei-Atweneboana[3,5]

**ABSTRACT** Genetic diversity in *Schistosoma mansoni* populations can impact the prevalence of resistance-conferring alleles, affecting treatment outcomes and control measures. Understanding how parasites are structured across and within populations is important to monitor and manage treatment impacts and the spread of drug resistance. This study assessed the genetic diversity and structure of *S. mansoni* populations under treatment using multi-locus microsatellite markers. Egg-positive stool samples from school-aged children in schistosomiasis-endemic communities were analyzed to obtain a binary matrix for GenAlEx 6.502 analysis. The markers were 78.57% polymorphic, with a mean allele number ranging from 2.50 to 4.75. Average population metrics were as follows: different alleles (Na) = 3.179, effective alleles (Ne) = 2.282, expected heterozygosity (He) = 0.442, and genetic identity (I) = 0.800, indicating high diversity. Diversity indices were particularly high in Tomefa (I = 0.931, He = 0.499), Manheam (I = 1.074, He = 0.557), and Adakope (I = 0.778, He = 0.448). Analysis of molecular variance revealed low differentiation among populations (4% of total variance, P = 0.001) and high differentiation among individuals (88% of total variance, P = 0.001), with $F_{ST}$ = 0.04 and high gene flow (Nm = 5.959). A neighbor-joining tree and principal coordinate analysis indicated low population structuring. The Mantel test showed no significant correlation between genetic and geographic distances (r = 0.006, P = 0.148). The findings suggest high genetic diversity and gene flow, supporting the potential spread of alleles that may confer traits like drug resistance. An integrated approach is needed to achieve sustainable control of schistosomiasis.

**IMPORTANCE** This study is crucial as it provides a detailed analysis of *Schistosoma mansoni* genetic diversity and population structure in schistosomiasis-endemic communities under treatment. By employing multi-locus microsatellite markers, the research highlights significant genetic variation within and among populations, revealing high levels of genetic diversity and gene flow, as well as the presence of private alleles. These findings emphasize the potential for the emergence and spread of drug resistance and virulence traits, which can impact treatment efficacy and control measures. The absence of significant geographic isolation in population structure further emphasizes the need for a comprehensive approach to schistosomiasis control, as genetic factors rather than geographic barriers may drive resistance dynamics. This study informs strategies for sustainable control by integrating genetic insights into disease management efforts.

**KEYWORDS** schistosomiasis, genetic diversity, hotspot, parasite, population

*S*chistosoma mansoni* causes human intestinal schistosomiasis in Africa, West Asia, part of South America, and the Caribbean region (1, 2). The control of these parasites is mainly focused on reducing morbidity through mass drug administration (MDA) to

**Peer Reviewer** Marcello Otake Sato, Niigata University of Pharmacy and Medical and Life Sciences, Niigata, Niigata, Japan

Address correspondence to Mike Yaw Osei-Atweneboana, oseiatweneboana@yahoo.co.uk.

The authors declare no conflict of interest.

school-age children and, if possible, pre-school-age children and, girls and women of reproductive age (3, 4).

As endemic countries make efforts toward control and elimination of neglected tropical diseases (NTDs), it has become necessary to track drug efficacy and genetic diversity of parasites, especially in areas where MDAs have been ongoing for at least a decade and where treatment coverages are high. This is critical for the early detection of drug resistance and changes in genetic diversity as well as the population structure of parasites (5–7). In Ghana, the annual MDA of praziquantel (PZQ) is very well established with high political motivation (5, 8). Implemented since 2008, however, several epidemiological studies conducted over the years have indicated that the disease is still prevalent in many parts of the country (8–10).

Genetic diversity of schistosome populations is important for analyzing the impact of PZQ treatment-induced selective pressures, which could lead to the emergence of new genotypes that are either tolerant or drug-resistant (11). The prolonged use of PZQ for treatment can potentially create strong selective pressure on these populations. This pressure could potentially favor the survival and propagation of *Schistosoma* genotypes that possess genetic mutations or traits conferring tolerance or resistance to the drug. Genetic subdivision of parasite populations (population structure) also provides relevant genetic information for assessing the potential emergence and spread of PZQ resistance (12). As drug pressure increases over time, there is the need to monitor the possible development of drug resistance, since there may be a selection of resistant phenotypes due to natural variation in the sensitivity of *Schistosoma* spp. to PZQ (13–15). Furthermore, studies have demonstrated that selection pressures can change the infectivity, virulence phenotypes, and population genetics of schistosomes within a few generations (11).

The epidemiology of schistosomiasis is affected by gene flow, effective population sizes, reproductive and mating systems, host transmission and specificity, as well as the standing variation of potential resistance alleles that influence the evolutionary potential of parasites to respond to drug pressures (16, 17). Due to the typically focal nature of schistosomiasis transmission, the population structure of *Schistosoma* species is expected to vary significantly between different geographic regions and epidemiological contexts (18). Therefore, perceptions of these spatial variations and their associated factors are important to help invent site-specific intervention strategies toward the control and elimination of the disease (19)

Various population genetic methods have been used for analyzing the genetic diversity of the parasite populations, including microsatellite markers and gene sequencing (20). Microsatellite markers are indirect methods, presumed to be neutral to drug resistance or immune evasion (21, 22). Previous studies have used multi-locus microsatellite analysis of individual schistosome miracidia obtained from infected human hosts to successfully characterize *S. mansoni* populations (21, 23). In some African countries including Egypt, Mali, Uganda, and Senegal, genetic variations of *S. mansoni* have been reported and linked to cure rate as well as disease severity (24, 25). However, there is limited information on the genetic diversity of schistosome populations in Ghana. In this study, we used multi-locus microsatellite markers to assess the genetic diversity and structure of *Schistosoma mansoni* populations across endemic communities in a small geographical area in Ghana.

## MATERIALS AND METHODS

### Study area

Four schistosomiasis-endemic communities—namely Tomefa (TOM), Manheam (MHM), Torgahkope (TGP), and Adakope (ADP)—were selected for this study (8). These communities are settlements located within the buffer zone of the Weija dam in the Ga South Municipal District, Ghana (Fig. 1). The district is mainly peri-urban, located about 17 km west of Accra (Ghana's capital city), and surrounds the Weija dam, a man-made

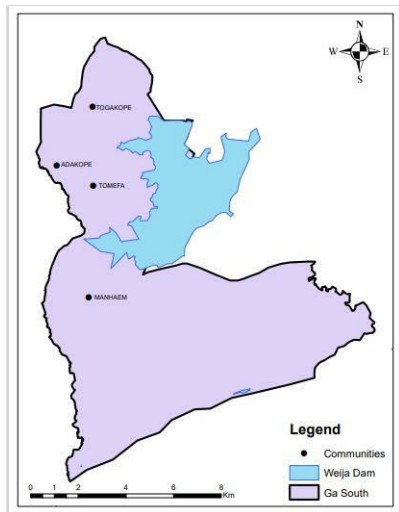

**FIG 1** Map showing the study communities located within the buffer zone of the Weija dam in the Ga South Municipal District. Map generated in ArcGIS (version 10.4.1).

reservoir created on the Densu river. The lake which supports the main water treatment plant supplying about 80% of potable water for the entire city of Accra and its environs for irrigation and fisheries programs has provided ideal conditions for disease transmission (26). Several studies on schistosomiasis since the creation of the Lake in 1979 show infections of both *S. mansoni* and *Schistosoma haematobium* among settlers in these areas (10, 26, 27). A recent study in this area reported *S. mansoni* prevalence ranging from 30% to 78% (8). Migration of disease-infected fishermen and peasant farmers has been found to be responsible for the increasing incidence of schistosomiasis in the lake basin (28). Other causes such as flaws in resettlement programs, changes in the flow rate of water, proliferation of water weeds, and conditions favoring the growth and development of host snails have all aided in the continuous transmission and infection after several rounds of school-based MDA in the district (8, 10, 26).

## Study population and sample selection

Samples used in this study were archived stool samples collected from school-aged participants (5–17 years old) of a larger epidemiological study investigating the impact of extended access to praziquantel and albendazole in whole communities (5, 8). Samples were collected at baseline, and at 2 and 6 months after a community-wide treatment (8) for parasitological analysis. Stool samples from study participants were subjected to Kato-Katz for parasite detection and quantification using two slides. Mean egg count was determined from the average of the egg counts from both slides prepared from a single stool sample. Infection intensity was reported as eggs per gram as recommended by the WHO. All community members, both study participants and non-participants, were treated with praziquantel using the standard dosage of 40 mg/kg under the observation of health workers from the district hospital. Treatment was done after baseline samples were collected and analyzed (8). For this study, only stool samples were selected for analysis. From each study community, a minimum of 30 *S. mansoni* egg-positive samples with egg intensity >100 eggs per gram of stool (EPG) were selected for genetic analyses at each time point.

## Genomic DNA extraction

Total genomic DNA (gDNA) was extracted from all selected samples, using Quick DNA Plus kits (Zymo Research, USA) and following the manufacturer's procedure with little modification for sample preparation. Briefly, *S. mansoni* eggs from each participant

were hatched to release miracidium. Single miracidium from each participant was transferred into a sterile 2 mL microcentrifuge tube containing 0.2 mm glass beads with 200 µL of 2% polyvinylpolypyrrolidone in 1× phosphate buffered saline. Samples were homogenized by vortexing using the MagNA Lyser (Hoffman-La Roche Ltd., Switzerland) to lyse cells and eliminate polyphenolic compounds that could interfere with polymerase action. The samples were then incubated overnight and treated with wash buffers I and II. DNA that bound to the silica membrane of the spin columns was eluted with 70 µL of elution buffer and stored at −20°C until further molecular analyses.

## Amplification of microsatellite markers

Seven polymorphic microsatellite markers (Table 1) for *S. mansoni*—*SMDA28*, *SMD25*, *SMD28*, *SMD89*, *SMU31768*, *CA11-1*, and *SMS9-1* (21)—were used to determine the genetic diversity within each study population. Genomic DNA from each sample was amplified in separate reactions with each primer set in a total reaction volume of 10 µL comprising 1× OneTaq master mix (MgCl$_2$, OneTaq DNA polymerase, and dNTPs [New England Biolabs Inc., UK]), 200 nM each of forward primer and reverse primer, and 3 µL of gDNA. PCR was performed in a thermal cycler, with the following amplification conditions: 45 cycles of denaturation at 94°C for 45 s, annealing at Tm (°C) (Table 1) for 45 s, and extension at 72°C for 45 s after an initial denaturation step of 94°C for 3 min, and a single cycle at 72°C for 5 min to end the reaction. PCR products were run in 1× Tris-acetate-EDTA (TAE) buffer on a 3% agarose gel stained with ethidium bromide. Gels were visualized under UV light from the transilluminator, and band patterns were scored from digital images of the products obtained.

## Microsatellite data analysis

Simple sequence repeat size-dependent fragments were scored by their molecular weight to obtain a binary matrix. Band patterns within the size ranges of each primer set were scored for clear, strong, and reproducible bands. Data were entered on a Microsoft Excel sheet. Samples from each community were considered as a population, and a cohort of participant samples, positive across all time points in TOM and MHM, was used to compare the effect of treatment on the populations.

GenAlEx 6.502 software (29) was used to estimate the number (*N*) of polymorphic bands, the number of effective alleles (Ne, the number of equally frequent alleles that would take to achieve the same expected heterozygosity as in your studied population), observed heterozygosity (Ho, estimated from individual genotypes directly and depends on both the amount of genetic variation in the population and the level of inbreeding, which increases homozygosity), expected heterozygosity (He, a fundamental measure of

**TABLE 1** *Schistosoma mansoni* microsatellite markers (21)[a]

| Marker/locus | Primer sequence | Clone size | Melting temperature, Tm (°C) |
|---|---|---|---|
| SMDA28 | Forward: CATGATCTTAGCTCAGAGAGCC | 92–128 | 44.4 |
| | Reverse: AGCCAGTATAGCGTTGATCATC | | |
| SMD25 | Forward: GATTCCCAAGATTAATGCC | 272–312 | 44.4 |
| | Reverse: GCCATTAGATAATGTACGTG | | |
| SMD28 | Forward: CATCACCATCAATCACTC | 230–245 | 46.6 |
| | Reverse: TATTCACAGTAGTAGGCG | | |
| SMD89 | Forward: AGACTACTTTCATAGCCC | 138–169 | 46.6 |
| | Reverse: TTAAACCGAAGCGAGAAG | | |
| SMU31768 | Forward: TACAACTTCCATCACTTC | 179–247 | 44.4 |
| | Reverse: CCATAAGAAAGAAACCAC | | |
| CA11-1 | Forward: TTCAAAACCATGAGCAATAGATAC | 191–231 | 46.6 |
| | Reverse: CAACAAACAAGAAGGCTGATTAG | | |
| SMS9-1 | Forward: ATTACGATTGCACAGATACTTTTG | 178–208 | 50.4 |
| | Reverse: TTTCAGAAATTTGTTTCCTCCTC | | |

[a]Primer (forward and reverse) sequences, clone sizes, and the corresponding melting temperatures.

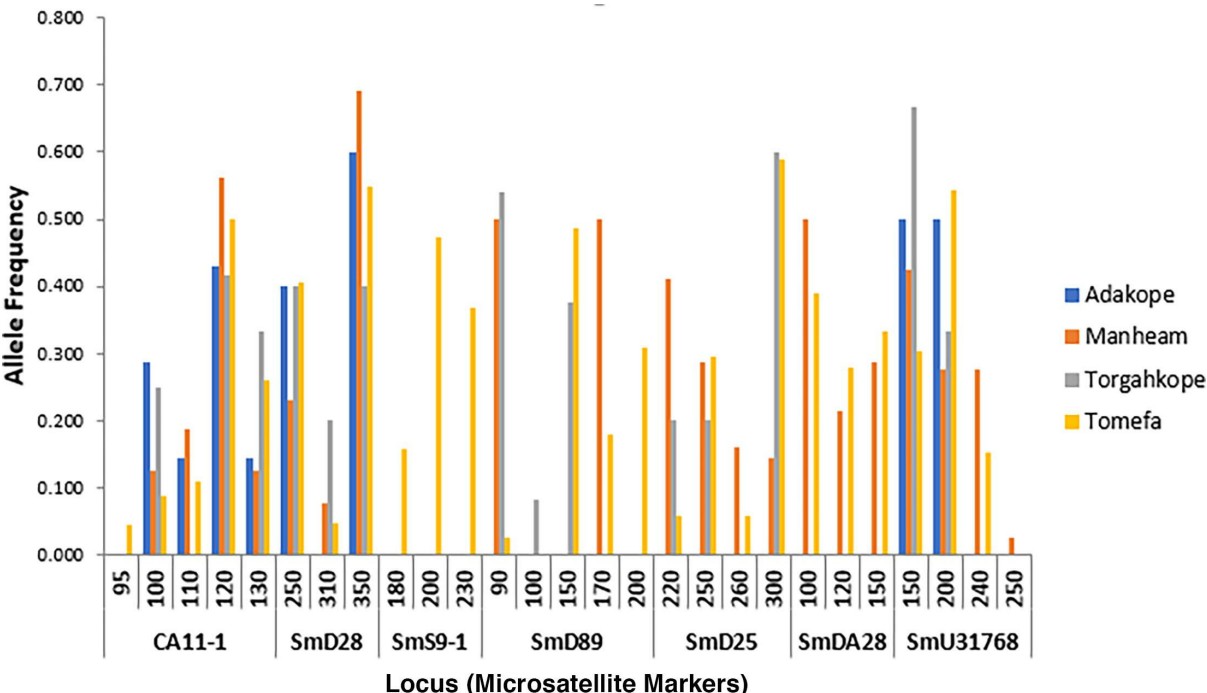

**FIG 2** Allelic frequency across populations. Number and frequency of alleles generated by each of the seven microsatellite markers across populations.

genetic variation in a population), Shannon's information index (*I*, a quantitative measure that reflects how many species there are in a community), *F* statistics ($F_{ST}$, the measure of population differentiation), Nei's matrix of genetic distance (*D*), and genetic identity (*I*) for each population. Based on Nei's *D*, a phylogenetic tree with bootstrap support values was constructed using MEGA 6 to illustrate the relationship between populations and the genetic population structure. GenAlEx 6.502 software was further used to compute the analysis of molecular variance (AMOVA) and pairwise fixation index ($F_{IS}$) as well as principal component analysis (PCoA) to evaluate the level of genetic differentiation among populations.

## RESULTS

### Marker variability

Out of the 240 individual stool samples analyzed from the four (30) communities, 136 were successfully genotyped by the seven microsatellite markers (*SmD25*, *SmDA28*, *SmD28*, *SmD89*, *SmU31768*, *CA11-1*, and *SmS9-1*) used. A total of 19 different alleles

**TABLE 2** Genetic parameters across populations[a]

| | Population | Na | Ne | I | Ho | He | F |
|---|---|---|---|---|---|---|---|
| MHM | Mean | 4.429 | 2.695 | 1.074 | 0.074 | 0.557 | 0.864 |
| | SE | 0.685 | 0.366 | 0.195 | 0.020 | 0.096 | 0.031 |
| ADP | Mean | 2.714 | 2.425 | 0.778 | 0.105 | 0.448 | 0.748 |
| | SE | 0.644 | 0.507 | 0.238 | 0.052 | 0.123 | 0.104 |
| TGP | Mean | 1.857 | 1.490 | 0.417 | 0.024 | 0.262 | 0.914 |
| | SE | 0.340 | 0.177 | 0.150 | 0.024 | 0.093 | 0.065 |
| TOM | Mean | 3.714 | 2.518 | 0.931 | 0.088 | 0.499 | 0.622 |
| | SE | 0.644 | 0.458 | 0.208 | 0.027 | 0.102 | 0.183 |
| Total | Mean | 3.179 | 2.282 | 0.800 | 0.073 | 0.442 | 0.770 |
| | SE | 0.337 | 0.208 | 0.106 | 0.017 | 0.054 | 0.060 |

[a]Na, number of different alleles; Ne, number of effective alleles; *I*, Shannon's information index; He, expected heterozygosity or gene diversity; Ho, observed heterozygosity; *F*, fixation index.

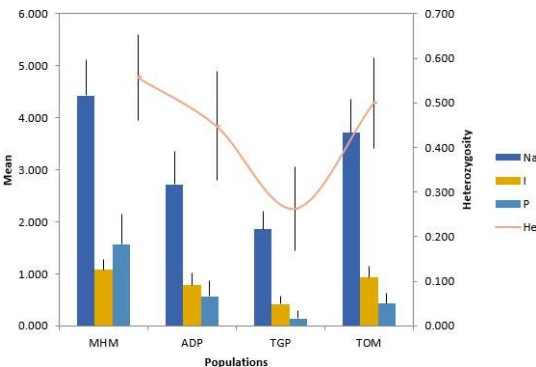

**FIG 3** Allelic patterns across populations. Na, number of different alleles; Ne, number of effective alleles; *I*, Shannon's information index; He, expected heterozygosity or gene diversity; P, private alleles.

(Na) were counted for all loci, yielding 300 alleles with significantly ($F = 26.179$, $P = 2.08 \times 10^{-09}$) varying numbers for each locus, ranging from 16 (*SmDA28*) to 52 (*SmD89*, *CA11-1*) across all populations (Fig. 2). All markers were 78.57% polymorphic with mean expected heterozygosity (He) ranging between 0.232 (*Ca11-1*) and 0.642 (*SmD89*) across all populations. A significant level of differentiation was observed across all loci ($P < 0.001$), indicating the efficiency of the markers in distinguishing between the populations.

## Population-level diversity and differentiation

The mean number of alleles (Na) was highest (Na = 4.429) in the TOM and MHM populations and lowest (Na = 1.857) in the ADP populations. Expected heterozygosity (He) ranged from 0.557 in MHM to 0.262 in TGP populations, while the observed heterozygosity ranged between ADP (0.105) and TGP (0.024) populations (Table 2), representing moderate to relatively high diversity across populations. There was a significant difference between expected and observed heterozygosity (Ho) ($x^2 = 0.27236$, df = 1, $P = 0.0014$) across all populations.

Consequently, the MHM population had the highest Shannon's information index (*I* = 1.094) followed by TOM (*I* = 0.931) and TGP (*I* = 0.778) with ADP recording the lowest (*I* = 0.417), confirming high diversity at least, in three populations. Fixative index (*F*) ranged from the highest (*F* = 0.914) in TGP, followed by MHM (*F* = 0.864) and ADP (*F* = 0.748) to the lowest (*F* = 0.622) in the TOM population (Table 2).

Private alleles (P) were found in all four populations at frequencies higher than 5%, most of which occurred in the MHM population (Fig. 3).

AMOVA based on inputs as allelic distance matrix for *F* statistics analysis showed significantly high variation ($P = 0.001$) among individuals (88%), but with low variation within individuals (8%) and among populations (4%) (Fig. 4). Based on standard permutation (999) across the full data set, $F_{ST}$, $F_{IS}$, $F_{IT}$, and Nm were 0.040, 0.918, 0.921, and 5.959, respectively, indicating low population differentiation, high inbreeding, and high gene flow.

## Phylogeography

Genetic distance (Nei's *D*) analysis revealed variations among the populations, with the highest distance observed between ADP and TOM (1.257) and the lowest between TGP and TOM (0.447). Nei's *I* values, representing genetic identity, ranged from 0.285 (lowest between ADP and TOM) to 0.639 (highest between TGP and TOM) (Table 3). The neighbor-joining tree depicted two main clusters, with TOM and TGP populations forming one cluster and MHM and ADP in separate clades (Fig. 5), highlighting distinct genetic relationships among the populations.

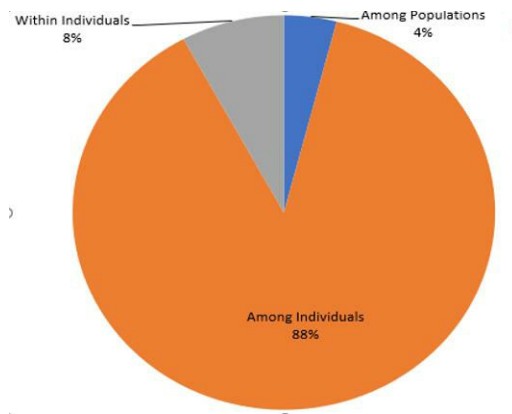

**FIG 4** Percentages of molecular variance across populations.

## Population structure

PCoA demonstrated a low level of population structuring, with axes 1, 2, and 3 explaining 58.92%, 24.62%, and 16.47% of the variation, respectively (Fig. 6). Despite significant genetic differentiation ($P = 0.036$), the standardized Mantel test did not reveal a significant correlation ($r = 0.006$, $P = 0.148$) between genetic and geographic distances, suggesting that the observed population structure is not solely influenced by geographic isolation. Pairwise $F_{ST}$ values indicated significant genetic differentiation among populations, ranging from 0.181 (lowest between MHM and TOM) to 0.392 (highest between ADP and TGP), possibly attributed to the family sub-structure (Table 3). Conversely, pairwise gene flow coefficients (Nm) suggested high gene flow between populations, with values ranging from 0.388 (between ADP and TGP) to 1.131 (between MHM and TOM) (Table 3).

## Diversity across treatment time points

Analysis at this level was run on a subset of the TOM and MHM population who were positive at all survey time points of the study. The parameters measured included Na = number of different alleles, Ne = number of effective alleles, He = expected heterozygosity or gene diversity, and number of private alleles. Measuring Na, Ne, He, and the number of private alleles together provides a comprehensive understanding of genetic diversity. Na and Ne indicate the range and distribution of alleles, He reflects the overall genetic variability within the population, and private alleles reveal unique genetic features. Together, these parameters offer a detailed view of genetic diversity and potential impacts on traits like drug resistance.

For the individuals from MHM, He reduced to 0 (zero) at the 2-month time point from 0.143 at baseline and increased to 0.254 at 6 months. This trend was similar to the Na and Ne parameters (Fig. 7). Private alleles were present at baseline and at 6 months with frequencies of 0.571 and 0.741, respectively (Fig. 7A).

The expected heterozygosity (He) of individuals from TOM also took a similar pattern, reducing to 0.141 at 2 months from 0.159 recorded at baseline and increasing to 0.319 at

**TABLE 3** Pairwise population Nei's genetic distance (*D*) and Nei's genetic identity (*I*) values, genetic differentiation coefficient ($F_{ST}$) and estimates of gene flow (Nm)

| Population 1 | Population 2 | Nei's *D* | Nei's *I* | $F_{ST}$ | Nm |
|---|---|---|---|---|---|
| MHM | ADP | 1.078 | 0.340 | 0.257 | 0.724 |
| MHM | TGP | 0.494 | 0.610 | 0.220 | 0.886 |
| ADP | TGP | 1.113 | 0.328 | 0.392 | 0.388 |
| MHM | TOM | 0.631 | 0.532 | 0.181 | 1.131 |
| ADP | TOM | 1.257 | 0.285 | 0.327 | 0.514 |
| TGP | TOM | 0.447 | 0.639 | 0.213 | 0.925 |

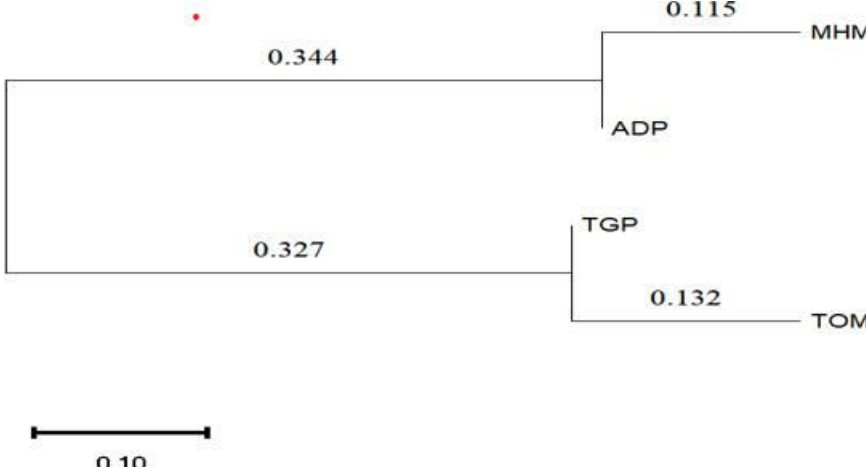

FIG 5  A neighbor-joining tree of the populations. The evolutionary history was inferred using the neighbor-joining method. The optimal tree is shown. The tree is drawn to scale, with branch lengths (above the branches) in the same units as those of the evolutionary distances used to infer the phylogenetic tree. Evolutionary analyses were conducted in MEGA X (31).

the 6-month time point (Fig. 7B). The Na and Ne, however, maintained a steady increase from the baseline (Na 0.857, Ne 0.829) to the 6-month time point (Na 1.714, Ne 1.632), and private alleles were present throughout the timelines with increasing frequencies (Fig. 7B).

A total of 11 different alleles—90, 95, 100, 110, 130, 150, 180, 200, 250, 310, and 350—were recorded within the MHM population at different frequencies over all time points. At baseline, seven alleles (90, 95, 100, 150, 200, 250, and 350) were present, while eight alleles were present at the 6-month time points. Allele 150 was present at all time points at the same frequency (Fig. 8A).

Twelve alleles (110, 120, 130, 150, 170, 200, 220, 230, 240, 260, 300, and 350) were recorded within the TOM population over the time points at different frequencies. At baseline, six alleles (100, 150, 170, 200, 240, and 350) were present. This increased to eight alleles at the 2-month time point (100, 120, 150, 200, 220, 240, 300, and 350). All 12 alleles were present at the 6-month time point. Five alleles were present at all survey time points (100, 150, 200, 240, and 350) (Fig. 8B).

We also looked at the mean allele frequency of the populations in relation to changes in mean egg count (EPG) across time points. In the MHM population, as the average EPG decreased sharply to 105 EPG at 2 months after treatment from 189 EPG before treatment and increased again to 321 EPG at 6 months post-treatment, the mean allele frequency also decreased significantly ($P = 0.031$) in a similar manner (Table 4). For the TOM population, although the EPG increased at 6 months post-treatment assessment

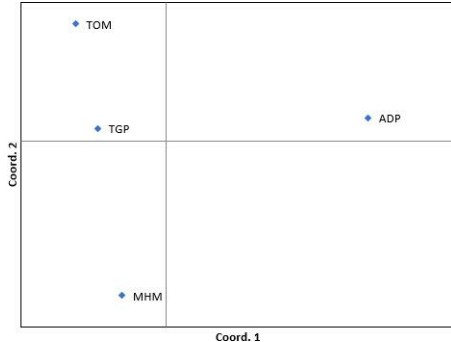

FIG 6  PCoA based on genetic distances.

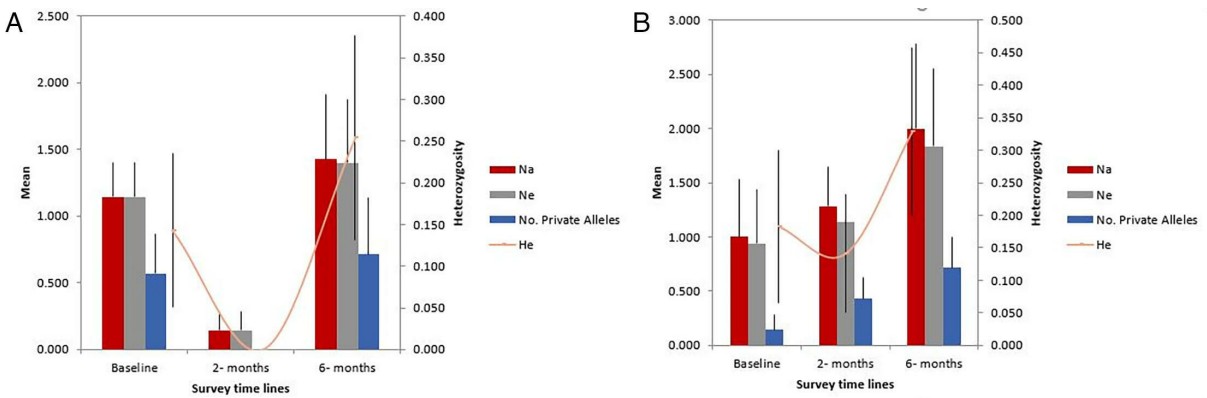

**FIG 7** Genetic parameters across time points for each population. MHM (A) and TOM (B).

after the sharp decrease at 2 months post-treatment, the mean allele frequency also decreased, although not significant ($P = 0.056$).

## DISCUSSION

Populations of *Schistosoma mansoni* exhibited substantial genetic diversity, as evidenced by many alleles, a high Shannon's information index, and elevated expected heterozygosity values across the loci analyzed. This level of genetic variation is particularly notable given the study's limited geographic scope. Each of the seven microsatellite markers employed generated at least three alleles and demonstrated high expected heterozygosity, indicating a significant level of polymorphism. This polymorphism highlights the markers' capacity to effectively differentiate between populations. These findings are consistent with previous studies investigating *S. mansoni* infra-populations among school children in Kenya, where similarly high levels of genetic diversity were observed (2, 32).

High levels of genetic diversity in our study populations can be attributable to several factors, key among them being co-infection of multiple genotypes from genetically different cercariae hypothesized to generate distinct schistosome lineages (32, 33). These four transmission sites had children who have been repeatedly exposed to *S. mansoni* infection (8, 10, 26, 27). This genetic diversity observed in the *S. mansoni* population can increase the resilience and adaptability of the parasite populations to environmental changes, including changes in host populations, treatment regimes, and ecological conditions. The significant genetic variation between these populations, as indicated by $F_{ST}$ values, further supports the notion of high genetic diversity within the *S. mansoni* populations. However, the calculated $F_{IS}$ values were positive, which is suggestive of a weak family structure and low level of genetic differentiation due to high levels of gene flow within populations (34).

The PCoA confirmed some variations among all populations, especially between the TOM and ADP populations, like findings made in a Kenyan study (35). This was also evident in the phylogeny tree which showed the populations on the extreme ends of different clades, although they are closer to each other geographically (Fig. 1). Again, the lowest genetic distance was between TGP and TOM populations, which were found in the same clade of the phylogeny tree, although they are the farthest apart geographically. There was no significant correlation between genetic differentiation and geographical distance; thus, these populations do not show a pattern of isolation by distance (32, 36). The findings suggest that human hosts in these transmission sites act as reservoirs and dispersal agents for *S. mansoni* genotypes, warranting further investigation in other populations.

While most alleles observed in this study were shared across all four populations, private alleles were detected in each population at frequencies exceeding 5%. The

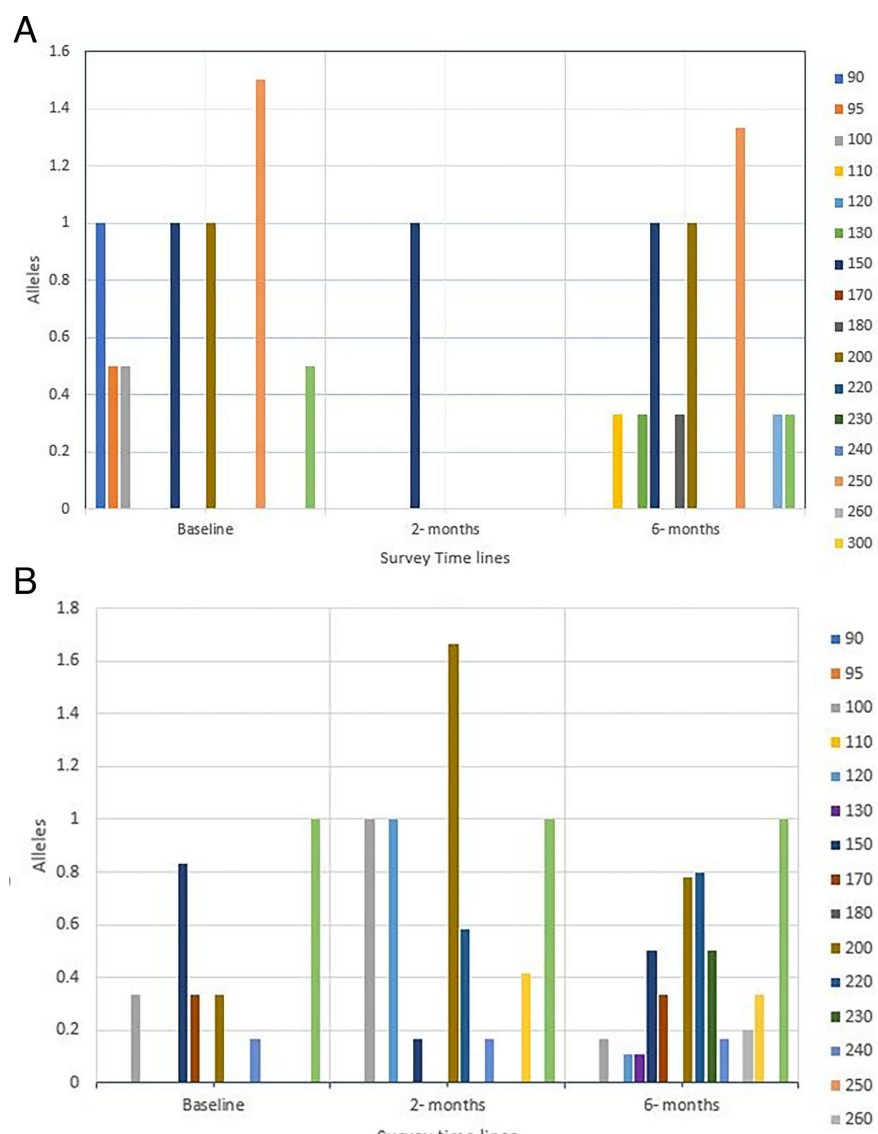

FIG 8   (A) Allele frequencies across time points for each population at MHM. (B) Allele frequencies across time points for each population at TOM.

presence of these private alleles could indicate exposure to unique *Schistosoma mansoni* genotypes specific to a single infection site (12). Alternatively, this pattern may result from a few highly fecund worm pairs with unique alleles contributing disproportionately to the offspring population, as the genetic material analyzed was derived from eggs rather than adult worms (35, 37). There could also be the possibility that different definitive hosts might be harboring different parasite strains in the population (18). In all four study communities, the observed heterozygosity was lower than expected, an indication of inbreeding. This relatively low heterozygosity could be due to our

TABLE 4   Total allele frequencies and mean egg count per population

| Timeline | MHM | | TOM | |
| | Allelic frequency | EPG | Allelic frequency | EPG |
|---|---|---|---|---|
| Baseline | 0.43 | 189 | 0.29 | 1,950 |
| 2 months | 0.07 | 105 | 0.26 | 468 |
| 6 months | 0.36 | 321 | 0.26 | 1,032 |

study design where samples from different individuals were combined to represent a single population and led to the Wahlund effect. The results, however, support the idea that school children move freely between the study sites which have several common infection foci (38, 39). An increase in heterozygosity and effective alleles over treatment time suggests a potential for novel resistance mutations. Gene flow between populations may also contribute to the spread of rare alleles, enhancing the risk of drug resistance emergence.

The high genetic diversity within populations may lead to drug resistance, despite a low probability (40). We saw little or no genetic differentiation between villages or between groups of villages with the same treatment intensity, but slightly higher genetic diversity within the pre-treatment compared to post-treatment parasite populations. The differential selection observed in post-treatment and pre-treatment parasite populations could be linked to any reduced susceptibility of parasites to praziquantel treatment. Studies conducted elsewhere have also indicated the increased chance of drug resistance in the presence of high genetic diversity (41, 42). The size ratio of treated to untreated populations, known as "refugium," is crucial in determining the rate of drug resistance allele occurrence in a population (35, 43). Our observations suggest that the refugium in our study is relatively large due to the lack of population sub-structure throughout the four transmission sites.

The prevalence of infection is predicted to rapidly rise to pre-control levels following MDA because of the non-linear relationship between prevalence and intensity (29). Our results showed that several alleles persisted throughout treatment, although there was a reduction in the intensity of infection in these populations. For example, in the TOM community, the 150 allele was present from the baseline at a frequency of 1.00 and reappeared at 2 and 6 months at the same frequency. The correlation between allele frequency and egg intensity was significant ($r = 0.8324$, $P = 0.0120$) across time points. This pattern may be a result of acquired resistance after repeated drug treatment or behavioral change, which should be investigated for successful monitoring and modeling of MDA strategies (28).

Allele counts were similar over treatment time points in the TOM population. At 2-month and 6-month time points, allele frequency was the same (0.26), but the intensity of infection increased by over twofold from 468.92 to 1032.9 EPG over this period. Similar results were reported in a previous study (33) which found that while EPG varied among patients by more than fourfold, parasite allele count was much more similar in the population. Additional studies are, however, needed to determine how allele counts correlate with worm burdens to help guide and monitor the treatment of individual patients or allow for the focus of treatment to specific geographic areas.

## Conclusion

The study highlights the significant characteristics of genetic diversity within *Schistosoma mansoni* populations, revealing substantial differentiation among individuals, high gene flow across populations, and the presence of private alleles over treatment time points. These findings suggest an environment conducive to the spread of rare alleles, potentially conferring traits such as drug resistance and virulence. High differentiation among individuals facilitates the emergence of new traits through natural selection, while high gene flow enables the rapid dissemination of advantageous alleles across different regions. The persistence of private alleles within specific populations further contributes to the overall genetic diversity and adaptive potential of *S. mansoni*. These findings emphasize the importance of monitoring genetic diversity to anticipate and address the emergence of drug resistance and other adaptive traits, informing more effective control and treatment strategies for schistosomiasis. It is worth mentioning that an ongoing schistosomiasis control program involving mass treatment of the whole community with praziquantel is providing a platform for more research into resistance genes within these communities.

## ACKNOWLEDGMENTS

The authors wish to thank the members of the four communities who participated in this study, especially the children who provided samples. Many thanks to the staff of CSIR who provided technical help during the study. Our sincere gratitude goes to Mr. Jeffrey Sumboh for generating the map of the study area.

COUNTDOWN (grant ID: PO6407 to LSTM) is a multi-disciplinary research consortium dedicated to investigating cost-effective, scaled-up, and sustainable solutions, necessary to control and eliminate the seven most common NTDs by 2020. COUNTDOWN was formed in 2014 and is funded by UKAID, part of the Department for International Development (DFID). The funders had no role in study design, data collection and analysis, decision to publish, or manuscript preparation.

Y.A.A., J.A., F.T.A., and M.Y.O-A. conceived and designed the study. Y.A.A., E.O.A., and F.T.A. performed the experiments, analyzed the data, and drafted the manuscript. I.O-F., L.B.D., R.L.D., S.A., and A.Y.D. participated in the implementation of the study. All authors read, reviewed, edited, and approved the final manuscript.

## AUTHOR AFFILIATIONS

[1]Department of Parasitology, Noguchi Memorial Institute for Medical Research, College of Health Sciences, University of Ghana, Accra, Ghana

[2]Department of Clinical Microbiology, Kwame Nkrumah University of Science and Technology, Kumasi, Ghana

[3]Biomedical and Public Health Research Unit, CSIR—Water Research Institute, Council for Scientific and Industry Research, Accra, Ghana

[4]Department of Medical Diagnostics, Kwame Nkrumah University of Science and Technology, Kumasi, Ghana

[5]CSIR College of Science and Technology, Accra, Ghana

## AUTHOR ORCIDs

Yvonne Aryeetey Ashong  http://orcid.org/0000-0003-3305-223X
Jewelna Akorli  http://orcid.org/0000-0002-3972-0860
Frank Twum Aboagye  http://orcid.org/0000-0002-3881-9789
Mike Yaw Osei-Atweneboana  http://orcid.org/0000-0002-8541-5015

## AUTHOR CONTRIBUTIONS

Yvonne Aryeetey Ashong, Conceptualization, Data curation, Formal analysis, Investigation, Methodology, Validation, Visualization, Writing – original draft, Writing – review and editing | Emmanuel Odartei Armah, Formal analysis, Investigation, Software, Writing – original draft, Writing – review and editing | Jewelna Akorli, Conceptualization, Data curation, Formal analysis, Investigation, Methodology, Validation, Visualization, Writing – original draft, Writing – review and editing | Frank Twum Aboagye, Data curation, Formal analysis, Investigation, Methodology, Software, Visualization, Writing – original draft, Writing – review and editing | Isaac Owusu-Frimpong, Writing – original draft, Writing – review and editing | Linda Batsa Debrah, Project administration, Writing – original draft, Writing – review and editing | Rhoda Lims Diyie, Writing – original draft, Writing – review and editing | Samuel Armoo, Project administration, Resources, Writing – review and editing | Alexander Yaw Debrah, Methodology, Project administration, Supervision, Validation, Writing – original draft, Writing – review and editing | Mike Yaw Osei-Atweneboana, Conceptualization, Funding acquisition, Supervision, Validation, Visualization, Writing – review and editing

## DATA AVAILABILITY

The datasets used and/or analyzed during the current study are available from the corresponding author upon reasonable request. The data has been deposited in a public repository, zenodo: https://doi.org/10.5281/zenodo.15767184

## ETHICS APPROVAL

The Institutional Review Board (IRB) of the Council for Scientific and Industrial Research (research protocol number 003/CSIR-IRB/2016) and the Liverpool School of Tropical Medicine (research protocol number 16-044) granted ethical clearance for this study. This study is embedded within a larger study on the expanded access to praziquantel and albendazole in whole communities, see references 7, 10 for full details on sensitisation, enrolment, and sampling procedures. Briefly, all participants were enrolled after community, household, and individual sensitisation. Prior to sample collection, all parents or guardians signed a written consent form, and all participating school-age children (12–17 years) assented to participate.

## ADDITIONAL FILES

The following material is available online.

### Open Peer Review

**PEER REVIEW HISTORY (review-history.pdf).** An accounting of the reviewer comments and feedback.

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
