## [Reviewer comments · Microbiology Spectrum]

Microbiology Spectrum

Evidence of high genetic diversity among parasite populations in a schistosomiasis hotspot

Yvonne Ashong, Emmanuel Armah, Jewelna Akorli, Frank Aboagye, Isaac Owusu-Frimpong, Linda Batsa Debrah, Rhoda Diyie, Samuel Armoo, Alexander Debrah, and Mike Osei-Atweneboana

Corresponding Author(s): Mike Osei-Atweneboana, Council for Scientific and Industrial Research

Review Timeline:

Submission Date:	September 23, 2024
Editorial Decision:	November 24, 2024
Revision Received:	January 15, 2025
Accepted:	May 24, 2025

Editor: Clinton Jones

Reviewer(s): Disclosure of reviewer identity is with reference to reviewer comments included in decision letter(s). The following individuals involved in review of your submission have agreed to reveal their identity: Marcello Otake Sato (Reviewer #1)

Transaction Report:

DOI: <https://doi.org/10.1128/spectrum.02272-24>

Re: Spectrum02272-24 (Evidence of high genetic diversity among parasite populations in a schistosomiasis hotspot)

Dear Ms. Yvonne Ashong:

Thank you for the privilege of reviewing your work. Below you will find my comments, instructions from the Spectrum editorial office, and the reviewer comments.

Revision Guidelines

Sincerely,
Clinton Jones
Editor
Microbiology Spectrum

Reviewer #1 (Comments for the Author):

The MS by Ashong and Cols. studied the populations of *Schistosoma* in an important endemic region of Ghana near Dakka, the Capital city using a microsatellite analysis approach. Interesting results were obtained, however, some points regarding the conclusions could be better explained.

Is the Weija dam the main infection site for the studied populations?

I couldn't see the microsatellite analysis by time, which could hint at the persistent patterns after treatment.
The results of the fecal examination appeared in the discussion, but it is not present in the results.
It is not clear how monitoring genetic diversity by microsatellite could predict resistance or direct treatment strategies.

Reviewer #2 (Comments for the Author):

An interesting study focussing on an important aspect of Schistosomiasis, helpful towards control and prevention.
Please see the attached manuscript with comments/edits embedded as tracked changes.

**Evidence of high genetic diversity among parasite populations in a**
**schistosomiasis hotspot**

**Yvonne Aryeetey Ashong^{1,2}, Emmanuel Odartei Armah³, Jewelna Efua Birago Akorli¹,**
**Frank Twum Aboagye³, Isaac Owusu-Frimpong³, Linda Batsa Debrah², Rhoda Lims**
**Diye³, Samuel Armoo³, Alexander Yaw Debrah⁴, Mike Yaw Osei-Atweneboana^{3,5*}.**

¹ Department of Parasitology, Noguchi Memorial Institute for Medical Research, College of
Health Sciences, University of Ghana, Legon, Ghana

² Department of Clinical Microbiology, Kwame Nkrumah University of Science and
Technology, Kumasi, Ghana.

³ Biomedical and Public Health Research Unit, CSIR – Water Research Institute, Council for
Scientific and Industry Research, Accra, Ghana.

⁴ Department of Medical Diagnostics, Kwame Nkrumah University of Science and
Technology, Kumasi, Ghana

⁵ CSIR-College of Science and Technology, Accra, Ghana

* **Correspondence:** oseiatweneboana@yahoo.co.uk

**Email Addresses**

YAA: yashong@noguchi.ug.edu.gh

EOA: emmanuelarmah44@yahoo.com

JEBA: jakorli@noguchi.ug.edu.gh

FTA: frankaboagye71@gmail.com

IOF: ki.owusufrimpong@gmail.com

LBD: lindrousy@yahoo.com

RLD: rho_lims@yahoo.com

SA: samuel.k.armoo@gmail.com

AYD: yadebrah@yahoo.com

**Abstract**

[revised manuscript text omitted]

polyvinylpyrrolidone (PVPP) in 1X Phosphate Buffered Saline (PBS). Samples were
homogenized by vortexing using the MagNA Lyser (Hoffman-La Roche Ltd., Switzerland) to
lyse cells and eliminate polyphenolic compounds that could interfere with polymerase action.
The samples were then incubated overnight and treated with Wash Buffers I and II. DNA that
bound to the silica membrane of the spin columns was eluted with 70 µL of Elution Buffer
and stored at -20⁰C until further molecular analyses.

**Amplification of microsatellite markers**

Seven polymorphic microsatellite markers (Additional file 2: Table S1) for *S. mansoni*;
*SMDA28*, *SMD25*, *SMD28*, *SMD89*, *SMU31768*, *CA11-1* and *SMS9-1* (24) were used to

determine the genetic diversity within each study population. Genomic DNA from each
sample was amplified in separate reactions with each primer set in a total reaction volume of
10 μ L comprising 1X OneTaq master mix [MgCl₂, OneTaq DNA polymerase and dNTPs
(New England Biolabs Inc., UK)], 200nM each of forward primer and reverse primer and
3 μ L of gDNA. PCR was performed in a thermal cycler, with the following amplification
conditions: 45 cycles of denaturation at 94°C for 45 sec, annealing at T_M°C (Table S1) for 45
178 sec, and extension at 72°C for 45 sec after an initial denaturation step of 94°C for 3mins, and a
179 single cycle at 72°C for 5 mins to end the reaction. PCR products were run in 1X TAE buffer
on a 3% agarose gel stained with ethidium bromide. Gels were visualized under UV light
from the transilluminator, and band patterns were scored from digital images of the products
obtained.

**Microsatellite Data Analysis**

Simple Sequence Repeats (SSR) size-dependent fragments were scored by their molecular
weight to obtain a binary matrix (**Fig. S1**). Band patterns within the size ranges of each
primer set were scored for clear, strong, and reproducible bands. Data was entered on a
Microsoft Excel sheet (**Additional file 3: Dataset 1**). Samples from each community were
considered as a population, and a cohort of participants samples, positive across all time
points in TOM and MHN were used to compare the effect of treatment on the populations.

GenAlEx 6.502 software (48) was used to estimate the number (N) of polymorphic bands, the
number of effective alleles (N_e, the number of equally frequent alleles that would take to
achieve the same expected heterozygosity as in your studied population), observed
heterozygosity (H_o, estimated from individual genotypes directly and depends on both the
amount of genetic variation in the population and the level of inbreeding, which increases

homozygosity), expected heterozygosity (H_e , a fundamental measure of genetic variation in a
population), Shannon's information index (I , a quantitative measure that reflects how many
species there are in a community), F- statistics (F_{ST} , the measure of population
differentiation), Nei's matrix of genetic distance (D) and genetic identity (I) for each
population. Based on Nei's D , a phylogenetic tree with bootstrap support values was
constructed using MEGA 6 to illustrate the relationship between populations and the genetic
population structure. GenAlEx 6.502 software was further used to compute the Analysis of
molecular variance (AMOVA) and Pairwise fixation index (F_{IS}) as well as Principal
Component Analysis (PCoA), to evaluate the level of genetic differentiation among
populations.

**Results**

**Marker variability**

Out of the 240 individual stool samples analysed from the four (4) communities, 136 were
successfully genotyped by the 7 microsatellite markers (*SmDA28*, *SmD28*, *SmD89*,
*SmU31768*, *CA11-1*, *SmS9-1*) used. A total of 19 different alleles (N_a) were counted for all
loci, yielding 300 alleles with significantly ($F= 26.179$, $p= 2.08 \times 10^{-09}$) varying numbers for
each locus, ranging from 16 (*SmDA28*) to 52 (*SmD89*, *CA11-1*) across all population (**Fig. 2**).
All markers were 78.57% polymorphic with mean expected heterozygosity (H_e) ranging
between 0.232 (*CA11-1*) and 0.642 (*SmD89*) across all populations. A significant level of
differentiation was observed across all loci ($p < 0.001$) indicating the efficiency of the
markers in distinguishing between the populations.

**Fig. 2: Allelic frequency across populations. Number and frequency of alleles generated**
 **by each of the seven microsatellite markers across population.**

**Population-level diversity and differentiation**

The mean number of alleles (N_a) was highest ($N_a = 4.429$) in the TOM and MHM
 populations and lowest ($N_a = 1.857$) in the ADP populations. Expected heterozygosity (H_e)
 ranged from 0.557 in MHM to 0.262 in TGP populations while the observed heterozygosity
 ranged between ADP (0.105) and TGP (0.024) populations (Table 1), representing moderate
 to relatively high diversity across populations. There was a significant difference between
 expected and observed heterozygosity (H_o) ($\chi^2 = 0.27236$, $df. = 1$, $p = 0.0014$) across all
 populations.

Consequently, the MHM population had the highest Shannon's Information Index ($I = 1.094$)
 followed by TOM ($I = 0.931$), and TGP ($I = 0.778$) with ADP recording the lowest ($I = 0.417$),
 confirming high diversity at least, in three populations. Fixative index (F) ranged from the

highest ($F = 0.914$) in TGP, followed by MHM ($F = 0.864$), ADP ($F = 0.748$) to the lowest
 (0.622) in the TOM populations (Table 1).

Table 1: Genetic parameters across populations

Population		Na	Ne	I	Ho	He	F
MHM	Mean	4.429	2.695	1.074	0.074	0.557	0.864
	SE	0.685	0.366	0.195	0.020	0.096	0.031
ADP	Mean	2.714	2.425	0.778	0.105	0.448	0.748
	SE	0.644	0.507	0.238	0.052	0.123	0.104
TGP	Mean	1.857	1.490	0.417	0.024	0.262	0.914
	SE	0.340	0.177	0.150	0.024	0.093	0.065
TOM	Mean	3.714	2.518	0.931	0.088	0.499	0.622
	SE	0.644	0.458	0.208	0.027	0.102	0.183
TOTAL	Mean	3.179	2.282	0.800	0.073	0.442	0.770
	SE	0.337	0.208	0.106	0.017	0.054	0.060

*Na* = number of different alleles; *Ne* = number of effective alleles; *I* = Shannon's
 Information Index; *He* = expected heterozygosity or gene diversity; *Ho* = observed
 heterozygosity; *F* = fixation index ...

Private alleles (P), were found in all four populations at frequencies higher than 5%, most of
 which occurred in the MHM population (Fig 3).

**Fig. 3: Allelic patterns across populations.**

[Na = number of different alleles; Ne = number of effective alleles; I = Shannon's
 Information Index; He = expected heterozygosity or gene diversity; P= Private alleles.]

Analysis of molecular variance (AMOVA) based on inputs as allelic distance matrix for F-

statistics analysis showed significantly high variation ($p = 0.001$) among individuals (88%),

but with low variation within individuals (8%) and among population (4%) (Fig. 4). Based on

standard permutation (999) across the full data set, F_{ST} , F_{IS} , F_{IT} and Nm were 0.040, 0.918,

0.921 and 5.959 respectively, indicating low population differentiation, high inbreeding and

high gene flow.

 **Fig. 4: Percentages of molecular variance across population**

**Phylogeography**

Genetic distance (Nei'D) analysis revealed variations among the populations, with the highest
 distance observed between ADP and TOM (1.257) and the lowest between TGP and TOM
 (0.447). Nei'I values, representing genetic identity, ranged from 0.285 (lowest between ADP
 and TOM) to 0.639 (highest between TGP and TOM) (Table 2). The neighbour-joining tree
 depicted two main clusters, with TOM and TGP populations forming one cluster and MHM
 and ADP in separate clades (Fig. 5), highlighting distinct genetic relationships among the
 populations.

Fig. 5: A neighbour-joining tree of the populations

[The evolutionary history was inferred using the Neighbour-Joining method. The optimal tree is shown. The tree is drawn to scale, with branch lengths (above the branches) in the same units as those of the evolutionary distances used to infer the phylogenetic tree. Evolutionary analyses were conducted in MEGA X (30).]

**Population Structure**

Principal Component Analysis (PCoA) demonstrated a low level of population structuring,
with axes 1, 2, and 3 explaining 58.92%, 24.62%, and 16.47% of the variation, respectively
(Fig. 6). Despite significant genetic differentiation ($p = 0.036$), the standardized Mantel test
did not reveal a significant correlation ($r=0.006$, $p=0.148$) between genetic and geographic
distances, suggesting that the observed population structure is not solely influenced by
geographic isolation. Pairwise F_{ST} values indicated significant genetic differentiation among
populations, ranging from 0.181 (lowest between MHM and TOM) to 0.392 (highest between
ADP and TGP), possibly attributed to family sub-structure (Table 2). Conversely, pairwise
gene flow coefficients (N_m) suggested high gene flow between populations, with values
ranging from 0.388 (between ADP and TGP) to 1.131 (between MHM and TOM) (Table 2).

**Fig. 6: Principal Component Analysis (PCA) based on genetic distances**

**Table 3. Pairwise Population Nei Genetic Distance (D) and Nei Genetic Identity (I)**
 **values; Genetic differentiation coefficient (Fst) and estimates of Geneflow (Nm)**

Population 1	Population 2	NEI D	NEI I	F _{ST}	NM
MHM	ADP	1.078	0.340	0.257	0.724
MHM	TGP	0.494	0.610	0.220	0.886
ADP	TGP	1.113	0.328	0.392	0.388
MHM	TOM	0.631	0.532	0.181	1.131
ADP	TOM	1.257	0.285	0.327	0.514
TGP	TOM	0.447	0.639	0.213	0.925

289

290 **Diversity across treatment time points**

291 Analysis at this level was run on a subset of the TOM and MHM population who were
 292 positive at all survey time point of the study. The parameters measured included N_a =

[revised manuscript text omitted]

	Reverse: GCCATTAGATAATGTACGTG		
SMD28	Forward: CATCACCATCAATCACTC	230–245	46.6
	Reverse: TATCACAGTAGTAGGCG		
SMD89	Forward: AGACTACTTTCATAGCCC	138–169	46.6
	Reverse: TTAAACCGAAGCGAGAAG		
SMU31768	Forward: TACAACCTCCATCACTTC	179–247	44.4
	Reverse: CCATAAGAAAGAAACCAC		
CA11-1	Forward: TTCAAAACCATGAGCAATAGATAC	191–231	46.6
	Reverse: CAACAAACAAGAAGGCTGATTAG		
SMS9-1	Forward: ATTACGATTGCACAGATACTTTTG	178–208	50.4
	Reverse: TTTCAGAAATTTGTTTCCTCCTC		

**Additional file 3: Dataset S1.** Excel spreadsheet of recorded clone sizes for each of the
 seven markers

**Abbreviations**

WHO: World Health Organization; MDA: Mass drug administration; SAC: school-age
children; PSAC: pre-school age children; PXQ: praziquantel; NTD: neglected tropical
diseases; TOM: Tomefa; MHM: Manheam; TGP: Torgahkope; ADP: Adakope; gDNA: Total
genomic DNA; PVPP: polyvinylpyrrolidone; PBS: phosphate buffered saline; SSR:
Simple Sequence Repeats; N: the number of polymorphic bands; Ne: the number of effective
alleles; Ho: observed heterozygosity; He: expected heterozygosity; I: Shannon's information
index; Nei's D: Nei's matrix of genetic distance for each population; Nei's I: matrix of
genetic identity for each population; AMOVA: Analysis of molecular variance; F_{IS} : Pairwise
fixation index; PCoA: Principal Component Analysis; EPG: egg per gram of stool.

**Declarations**

**Ethics approval and consent to participate**

The Institutional Review Board (IRB) of the Council for Scientific and Industrial Research
(research protocol number 003/CSIR-IRB/2016) and the Liverpool School of Tropical
Medicine (research protocol number 16-044) granted ethical clearance for this study. This
study is embedded within a larger study on the expanded access to praziquantel and
albendazole in whole communities, see (7,10) for full details of sensitisation, enrolment and
sampling procedures. Briefly, all participants were enrolled after community, household and
individual sensitisation. Prior to sample collection, all parents or guardians signed a written
consent form, and all participating school-age children (12-17 years) assented to participate.

**Availability of data and materials**

The datasets used and/or analysed during the current study are available from the
corresponding author upon reasonable request.

**Competing interests**

The authors have declared that no competing interests exist

**Funding**

COUNTDOWN (grant ID is PO6407 to LSTM) is a multi-disciplinary research consortium
dedicated to investigating cost-effective, scaled-up and sustainable solutions, necessary to
control and eliminate the seven most common NTDs by 2020. COUNTDOWN was formed
in 2014 and is funded by the UKAID part of the Department for International Development
(DFID). The funders had no role in study design, data collection and analysis, decision to
publish, or manuscript preparation.

**Authors' contributions**

YAA, JEBA, FTA and MYO conceived and designed the study. YAA, EOA and FTA
performed the experiments, analysed the data, and drafted the manuscript. IOF, LBD, RLD,
SA and AYD participated in the implementation of the study. All authors read, reviewed,
edited and approved the final manuscript.

**Acknowledgements**

The authors wish to thank the members of the four communities which participated in this
study especially the children that provided samples. Many thanks to the staff of CSIR that

provided technical help during the study. Our sincere gratitude goes to Mr. Jeffrey Sumboh
for generating the map of the study area.

**Reference**

- 1. Adenowo AF, Oyinloye BE, Ogunyinka BI, Kappo AP. Impact of human
schistosomiasis in sub-Saharan Africa. *Brazilian Journal of Infectious Diseases*
[Internet]. 2015 Mar 1 [cited 2022 Mar 5];19(2):196–205. Available from:
<http://dx.doi.org/10.1016/j.bjid.2014.11.004>
- 2. Morgan JAT, Dejong RJ, Adeoye GO, Ansa EDO, Barbosa CÇS, Brémond P, et al.
Origin and diversification of the human parasite *Schistosoma mansoni*. *Mol Ecol*
[Internet]. 2005 Oct [cited 2022 May 20];14(12):3889–902. Available from:
<https://pubmed.ncbi.nlm.nih.gov/16202103/>
- 3. van der Werf MJ, de Vlas SJ, Brooker S, Looman CWN, Nagelkerke NJD, Dik J, et
al. Quantification of clinical morbidity associated with schistosome infection in sub-
Saharan Africa. *Acta Tropica* [Internet]. 2003 [cited 2020 Dec 21];86:125–39.
Available from: www.elsevier.com/locate/actatropica
- 4. Steinmann P, Keiser J, Bos R, Tanner M, Utzinger J. Schistosomiasis and water
resources development: systematic review, meta-analysis, and estimates of people at
risk. *Lancet Infect Dis*. 2006 Jul;6(7):411–25.
- 5. Webster JP, Neves MI, Webster BL, Pennance T, Rabone M, Gouvras AN, et al.
Parasite population genetic contributions to the schistosomiasis consortium for
operational research and evaluation within sub-Saharan Africa. *American Journal of*
*Tropical Medicine and Hygiene*. 2020 Mar 1; 103: 80–91.

- 6. Colley DG, Bustinduy AL, Secor WE, King CH. Human schistosomiasis. In: The
Lancet. Elsevier B.V.; 2014. p. 2253–64.
- 7. Campbell SJ, Osei-Atweneboana MY, Stothard R, Koukounari A, Cunningham L,
Armoo SK, et al. The Countdown study protocol for expansion of mass drug
administration strategies against schistosomiasis and soil-transmitted helminthiasis in
Ghana. *Tropical Medicine and Infectious Disease*. 2018 Jan 22;3(1).
- 8. Casulli A. New global targets for NTDs in the WHO roadmap 2021–2030. *PLOS*
*Neglected Tropical Diseases* [Internet]. 2021 May 1 [cited 2022 Apr 22];15(5):
e0009373. Available from:
<https://journals.plos.org/plosntds/article?id=10.1371/journal.pntd.0009373>
- 9. Li EY, Gurarie D, Lo NC, Zhu X, King CH. Improving public health control of
schistosomiasis with a modified WHO strategy: a model-based comparison study. *The*
*Lancet Global Health*. 2019 Oct 1;7(10): e1414–22.
- 10. Cunningham LJ, Campbell SJ, Armoo S, Koukounari A, Watson V, Selormey P, et al.
Assessing expanded community-wide treatment for schistosomiasis: Baseline
infection status and self-reported risk factors in three communities from the Greater
Accra Region, Ghana. *PLoS Neglected Tropical Diseases*. 2020 Apr 1;14(4):1–22.
- 11. Lodh N, Naples JM, Bosompem KM, Quartey J, Shiff CJ. Detection of Parasite-
Specific DNA in Urine Sediment Obtained by Filtration Differentiates between Single
and Mixed Infections of *Schistosoma mansoni* and *S. haematobium* from Endemic
Areas in Ghana. *PLOS ONE* [Internet]. 2014 Mar 14 [cited 2022 Feb 16];9(3):
e91144. Available from:
<https://journals.plos.org/plosone/article?id=10.1371/journal.pone.0091144>

- 12. Anyan WK, Abonie SD, Aboagye-Antwi F, Tettey MD, Nartey LK, Hanington PC, et
al. Concurrent *Schistosoma mansoni* and *Schistosoma haematobium* infections in a
peri-urban community along the Weija dam in Ghana: A wake-up call for effective
National Control Programme. *Acta Tropica* [Internet]. 2019 Nov 1 [cited 2019 Aug
16]; 199:105116. <https://linkinghub.elsevier.com/retrieve/pii/S0001706X1831163X>
- 13. Webster JP, Molyneux DH, Hotez PJ, Fenwick A. The contribution of mass drug
administration to global health: Past, present and future. Vol. 369, *Philosophical*
*Transactions of the Royal Society B: Biological Sciences*. Royal Society of London;
2014.
- 14. Werkman M, Wright JE, Truscott JE, Oswald WE, Halliday KE, Papaiakovou M, et
al. The impact of community-wide, mass drug administration on aggregation of soil-
transmitted helminth infection in human host populations. *Parasites and Vectors*
[Internet]. 2020 Jun 8 [cited 2022 Mar 31];13(1):1–12. Available from:
<https://parasitesandvectors.biomedcentral.com/articles/10.1186/s13071-020-04149-4>
- 15. El-Kady AM, EL-Amir MI, Hassan MH, Allemailem KS, Almatroudi A, Ahmad AA.
Genetic Diversity of *Schistosoma haematobium* in Qena Governorate, Upper Egypt.
*Infection and Drug Resistance* [Internet]. 2020 Oct 15 [cited 2020 Nov 12]; Volume
13:3601–11. Available from: [https://www.dovepress.com/genetic-diversity-of-](https://www.dovepress.com/genetic-diversity-of-schistosoma-haematobium-in-qena-governorate-upper-peer-reviewed-article-IDR)
[schistosoma-haematobium-in-qena-governorate-upper-peer-reviewed-article-IDR](https://www.dovepress.com/genetic-diversity-of-schistosoma-haematobium-in-qena-governorate-upper-peer-reviewed-article-IDR)
- 16. Gower CM, Gabrielli AF, Sacko M, Dembelé R, Golan R. Population genetics of
*Schistosoma haematobium*: development of novel microsatellite markers and their
application to schistosomiasis control in Mali. *Parasitology*. 2011; 138:978–94.

- 17. Coeli R, Baba EH, Araujo N, Coelho PMZ, Oliveira G. Praziquantel Treatment
Decreases *Schistosoma mansoni* Genetic Diversity in Experimental Infections. PLoS
Neglected Tropical Diseases. 2013;7(12).
- 18. Berger DJ, Crellen T, Lamberton PHL, Allan F, Tracey A, Noonan JD, et al. Whole-
genome sequencing of *Schistosoma mansoni* reveals extensive diversity with limited
selection despite mass drug administration. Nature Communications [Internet]. 2021
Dec 1 [cited 2022 Jan 31];12(1). Available from: /pmc/articles/PMC8346512/
- 19. Lamberton PHL, Faust CL, Webster JP. Praziquantel decreases fecundity in
*Schistosoma mansoni* adult worms that survive treatment: Evidence from a laboratory
life-history trade-offs selection study. Infectious Diseases of Poverty [Internet]. 2017
Jun 16 [cited 2022 Apr 4];6(1):1–11. Available from:
<https://idpjournal.biomedcentral.com/articles/10.1186/s40249-017-0324-0>
- 20. Gibson AK. Genetic diversity and disease: The past, present, and future of an old
idea. 2021; <https://doi.org/10.1111/evo.14395>
- 21. Steinauer ML, Christie MR, Blouin MS, Agola LE, Mwangi IN, Maina GM, et al.
Non-Invasive Sampling of Schistosomes from Humans Requires Correcting for
Family Structure. PLOS Neglected Tropical Diseases [Internet]. 2013 [cited 2022 Apr
21];7(9): e2456. Available from:
- 22. Zhou Y biao B, Liang S, Chen G xin X, Rea C, Han S min M, He Z gui G, et al.
Spatial-temporal variations of *Schistosoma japonicum* distribution after an integrated
national control strategy: a cohort in a marshland area of China. BMC Public Health
[Internet]. 2013;13(1):1. Available from: BMC Public Health

- 23. Rollinson D, Webster JP, Webster B, Nyakaana S, Jørgensen A, Stothard JR. Genetic
diversity of schistosomes and snails: implications for control. *Parasitology*.
2009;136(13):1801–11.
- 24. GOWER CM, Shrivastava J, Lamberton PHELL, Rollinson D, Webster BL, Emery A,
et al. Development and application of an ethically and epidemiologically
advantageous assay for the multi-locus microsatellite analysis of *Schistosoma*
*mansoni*. *Parasitology* [Internet]. 2007 Apr 13 [cited 2019 Aug 20];134(Pt 4):523–36.
Available from:
<https://www.cambridge.org/core/product/identifier/S0031182006001685/type/journal>
[_article](https://www.cambridge.org/core/product/identifier/S0031182006001685/type/journal)
- 25. Figan CE, Sá JM, Mu J, Melendez-Muniz VA, Liu CH, Wellems TE. A set of
microsatellite markers to differentiate *Plasmodium falciparum* progeny of four genetic
crosses. *Malaria Journal* 2018 17:1 [Internet]. 2018 Feb 2 [cited 2022 Feb
15];17(1):1–6. Available from:
<https://malariajournal.biomedcentral.com/articles/10.1186/s12936-018-2210-z>
- 26. Beltran S, Galinier R, Allienne JF, Boissier J. Cheap, rapid and efficient DNA
extraction method to perform multilocus microsatellite genotyping on all *Schistosoma*
*mansoni* stages. 2008;103(August):501–3.
- 27. Ezech C, Yin M, Li H, Zhang T, Xu B, Sacko M, et al. High genetic variability of
*Schistosoma haematobium* in Mali and Nigeria. *Korean Journal of Parasitology*. 2015
Feb 1;53(1):129–34.
- 28. Afifi MA, Jiman-Fatani AA, Al-hussainy NH, Al-rabia MW, Bogari AA. Genetic
diversity among natural populations of *Schistosoma haematobium* might contribute to
inconsistent virulence and diverse clinical outcomes. *Journal of Microscopy and*

- Ultrastructure [Internet]. 2016 [cited 2020 Nov 12];4(4):222. Available from:
<http://dx.doi.org/10.1016/j.jmau.2016.04.002>
- 29. Blanton RE. Population Structure and Dynamics of Helminthic Infection:
Schistosomiasis *. Microbiology Spectrum [Internet]. 2019 Jul 5 [cited 2019 Oct 25];
7(4). Available from:
<http://www.asmscience.org/content/journal/microbiolspec/10.1128/microbiolspec.AM>
E-0009-2019
- 30. Takezaki N, Nei M. Genetic distances and reconstruction of phylogenetic trees from
microsatellite DNA. Genetics. 1996 Sep;144(1):389-99.
doi: 10.1093/genetics/144.1.389. PMID: 8878702; PMCID: PMC1207511..
- 31. Agola LE, Mburu DN, DeJong RJ, Mungai BN, Muluvi GM, Njagi ENM, et al.
Microsatellite typing reveals strong genetic structure of *Schistosoma mansoni* from
localities in Kenya. Infect Genet Evol [Internet]. 2006 Nov [cited 2022 May
20];6(6):484–90. Available from: <https://pubmed.ncbi.nlm.nih.gov/16675308/>
- 32. Aemero M, Boissier J, Climent D, Moné H, Mouahid G, Berhe N, et al. Genetic
diversity, multiplicity of infection and population structure of *Schistosoma mansoni*
isolates from human hosts in Ethiopia. BMC Genetics. 2015 Dec 3;16(1).
- 33. Tetteh-Quarcoo PB, Attah SK, Donkor ES, Nyako M, Minamor AA, Afutu E, et al.
Urinary Schistosomiasis in Children—Still a Concern in Part of the Ghanaian Capital
City. Open Journal of Medical Microbiology. 2013;03(03):151–8.
- 34. Amoah LAO, Anyan WK, Aboagye-Antwi F, Abonie S, Tettey MD, Bosompem KM.
Environmental Factors and their Influence on Seasonal Variations of Schistosomiasis
Intermediate Snail Hosts Abundance in Weija Lake, Ghana. Journal of Advocacy,
Research and Education. 2017;4(2):68–80.

- 35. Durand P, Sire C, Théron A. Isolation of microsatellite markers in the digenetic
trematode *Schistosoma mansoni* from Guadeloupe island. 2000; c:997–8.
- 36. Danso- Appiah A, Garner P, Olliaro PL, Utzinger J. Treatment of urinary
schistosomiasis: methodological issues and research needs identified through a
Cochrane systematic review. *Parasitology* [Internet]. 2009 Nov [cited 2022 May
20];136(13):1837–49. Available from: <https://pubmed.ncbi.nlm.nih.gov/19493363/>
- 37. Curtis J, Minchella DJ. Schistosome Population Genetic Structure: When Clumping
Worms is not Just Splitting Hairs. *Parasitology Today*. 2000;16(2):68–71.
- 38. CURTIS J, SORENSEN RE, PAGE LK, MINCHELLA DJ. Microsatellite loci in the
human blood fluke *Schistosoma mansoni* and their utility for other schistosome
species. *Molecular Ecology Notes*. 2001; 1: 143–5.
- 39. Agola LE, Steinauer ML, Mburu DN, Mungai BN, Mwangi IN, Magoma GN, et al.
Genetic diversity and population structure of *Schistosoma mansoni* within human
infrapopulations in Mwea, central Kenya assessed by microsatellite markers. *Acta
Tropica*. 2009;111(3):219–25.
- 40. Thiele EA, Sorensen RE, Gazzinelli A, Minchella DJ. Genetic diversity and
population structuring of *Schistosoma mansoni* in a Brazilian village.
- 41. Rudge JW, Carabin H, Balolong E, Tallo V, Shrivastava J, Lu DB, et al. Population
genetics of *Schistosoma japonicum* within the Philippines suggest high levels of
transmission between humans and dogs. *PLoS Neglected Tropical Diseases*.
2008;2(11).
- 42. Rodrigues NB, Loverde PT, Romanha AJ, Oliveira G. Characterization of New
*Schistosoma mansoni* Microsatellite Loci in Sequences Obtained from Public DNA
Databases and Microsatellite Enriched Genomic Libraries. 2002; 97:71–5.

- 43. Stohler RA, Curtis J, Minchella DJ. A comparison of microsatellite polymorphism
and heterozygosity among field and laboratory populations of *Schistosoma mansoni*.
*Int J Parasitol* [Internet]. 2004 Apr [cited 2022 May 20];34(5):595–601. Available
from: <https://pubmed.ncbi.nlm.nih.gov/15064124/>
- 44. Steinauer ML, Hanelt B, Agola LE, Mkoji GM, Loker ES. Genetic structure of
*Schistosoma mansoni* in western Kenya: The effects of geography and host sharing.
*International Journal for Parasitology*. 2009 Oct;39(12):1353–62.
- 45. Anderson R, Truscott J, Hollingsworth TD. The coverage and frequency of mass drug
administration required to eliminate persistent transmission of soil-transmitted
helminths. *Philosophical Transactions of the Royal Society B: Biological Sciences*.
2014 Jun 19;369(1645).
- 46. Lelo AE, Mburu DN, Magoma GN, Mungai BN, Kihara JH, Mwangi IN, et al. No
Apparent Reduction in Schistosome Burden or Genetic Diversity Following Four
Years of School-Based Mass Drug Administration in Mwea, Central Kenya, a Heavy
Transmission Area. *PLoS Neglected Tropical Diseases*. 2014;8(10).
- 47. Steinauer ML, Christie MR, Blouin MS, Agola LE, Mwangi IN, Maina GM, et al.
Non-Invasive Sampling of Schistosomes from Humans Requires Correcting for
Family Structure. *PLoS Neglected Tropical Diseases*. 2013;7(9).
- 48. Peakall R, Smouse PE. GenAEx 6.5: genetic analysis in Excel. Population genetic
software for teaching and research-an update. *Bioinformatics Applications* [Internet].
2012; 28: 2537–9. Available from www.eeb.uconn.edu/people/plewis/software.php
- 49. Ampofo, J. A., & Zuta, P. C. (1995). Schistosomiasis in the Weija Lake: A case study
of the public health importance of man-made lakes. *Lakes & Reservoirs: Research &*
*Management*, 1(3), 191-195.

1. **Reviewer Comment:** Is the Weija dam the main infection site for the studied populations?

Response

Thank you for this observation. The Weija Dam was identified as a primary site influencing transmission dynamics in the study area. This is due to its role as a water source for domestic and agricultural purposes, which increases human-water contact. We will clarify this in the manuscript, particularly in the study area section of the methods, by providing detailed information on the geographic scope of the sampling sites and their epidemiological significance on Line 115 to 130

2. **Reviewer Comment:** I couldn't see the microsatellite analysis by time, which could hint at the persistent patterns after treatment.

Response

We acknowledge the comment. However, we conducted a temporal analysis of genetic diversity across treatment time points for individuals in the TOM and MHM populations who tested positive throughout the study period. The detailed analysis is presented under the heading **diversity across timepoints** from line **282 to 308** and supported with figure 7.

Allele frequency cross the time points from base to six months post treatment is presented on **Line 311 to 318** and supported with **Figure 8A and Figure 8B**. We will highlight these findings in the revised manuscript

3. **Reviewer Comment:** The results of the faecal examination appeared in the discussion, but it is not present in the results.

Response

Thank you for highlighting this discrepancy. The results of faecal examination reporting the egg per gram (EPG) and the allele frequency is presented in **Table 3** and detailed on **Line 320 to 326**. We will ensure to highlight these results to improve the coherence between the results and discussion sections.

4. **Reviewer Comment:** It is not clear how monitoring genetic diversity by microsatellite could predict resistance or direct treatment strategies.

Response

This is an important point. Monitoring genetic diversity using microsatellites provides insight into population structure, gene flow, and potential selective pressures within parasite populations. High genetic diversity may indicate the potential for adaptation to selective pressures, such as drug treatment. We will expand the discussion to better articulate how this information can contribute to predicting resistance development and guiding targeted

treatment strategies. This will include referencing studies that have successfully linked genetic diversity metrics to resistance patterns.

Re: Spectrum02272-24R1 (Evidence of high genetic diversity among parasite populations in a schistosomiasis hotspot)

Dear Prof. Mike Yaw Osei-Atweneboana:

Your manuscript has been accepted, and I am forwarding it to the ASM production staff for publication. Your paper will first be checked to make sure all elements meet the technical requirements. ASM staff will contact you if anything needs to be revised before copyediting and production can begin. Otherwise, you will be notified when your proofs are ready to be viewed.

Sincerely,
Clinton Jones
Editor
Microbiology Spectrum

Reviewer #1 (Comments for the Author):

In this revised version, the authors addressed the comments and suggestions I have made. However, I am still skeptical regarding the direct evidence on the "resistant/virulent strains" once only the remaining alleles diversity after treatment do not prove their resistance, which should be done in another type of experiment such as the fecal egg reduction tests or others... I think it is better to refrain from a possible overstating of the results obtained in this interesting study.